# The Role of Immunohistochemistry as a Surrogate Marker in Molecular Subtyping and Classification of Bladder Cancer

**DOI:** 10.3390/diagnostics14222501

**Published:** 2024-11-08

**Authors:** Tatiana Cano Barbadilla, Martina Álvarez Pérez, Juan Daniel Prieto Cuadra, Mª Teresa Dawid de Vera, Fernando Alberca-del Arco, Isabel García Muñoz, Rocío Santos-Pérez de la Blanca, Bernardo Herrera-Imbroda, Elisa Matas-Rico, Mª Isabel Hierro Martín

**Affiliations:** 1Pathology Department, Juan Ramón Jiménez University Hospital (HJRJ), 21005 Huelva, Spain; tcbarbadilla@gmail.com; 2Department of Human Physiology, Human Histology, Pathology, and Sports Physical Education, University of Malaga (UMA), 29071 Málaga, Spain; dannielpriet@gmail.com (J.D.P.C.); tdawiddevera@gmail.com (M.T.D.d.V.); mir06isa@gmail.com (I.G.M.); misabel.hierro.sspa@juntadeandalucia.es (M.I.H.M.); 3Institute of Biomedical Research in Malaga (IBIMA-Plataforma BIONAND), 29590 Málaga, Spain; alberca.urologia@gmail.com (F.A.-d.A.); rsantospdb@gmail.com (R.S.-P.d.l.B.); ber.urologia@gmail.com (B.H.-I.); ematas@uma.es (E.M.-R.); 4Laboratory of Molecular Biology of Cancer (LBMC), Centre for Medical and Health Research, University of Malaga (UMA), 29010 Málaga, Spain; 5Pathologý Department, Hospital Universitario Virgen de la Victoria (HUVV), 29010 Málaga, Spain; 6Urology Department, Hospital Universitario Virgen de la Victoria (HUVV), 29010 Málaga, Spain; 7Genitourinary Alliance for Research and Development (GUARD Consortium), 29071 Málaga, Spain; 8Department of Surgical Specialties, Biochemistry, and Immunology, University of Malaga (UMA), 29071 Málaga, Spain; 9Department of Cell Biology, Genetics, and Physiology, University of Malaga (UMA), 29071 Málaga, Spain

**Keywords:** bladder cancer, immunohistochemistry, molecular subtypes, molecular classification

## Abstract

Background/Objectives: Bladder cancer (BC) is a highly heterogeneous disease, presenting clinical challenges, particularly in predicting patient outcomes and selecting effective treatments. Molecular subtyping has emerged as an essential tool for understanding the biological diversity of BC; however, its implementation in clinical practice remains limited due to the high costs and complexity of genomic techniques. This review examines the role of immunohistochemistry (IHC) as a surrogate marker for molecular subtyping in BC, highlighting its potential to bridge the gap between advanced molecular classifications and routine clinical application; Methods: We explore the evolution of taxonomic classification in BC, with a particular focus on cytokeratin (KRT) expression patterns in normal urothelium, which are key to identifying basal and luminal subtypes. Furthermore, we emphasise the need for consensus on IHC markers to reliably define these subtypes, facilitating wider and standardised clinical use. The review also analyses the application of IHC in both muscle-invasive (MIBC) and non-muscle-invasive bladder cancer (NMIBC), with particular attention to the less extensively studied NMIBC cases. We discuss the practical advantages of IHC for subtyping, including its cost effectiveness and feasibility in standard pathology laboratories, alongside ongoing challenges such as the requirement for standardised protocols and external validation across diverse clinical settings; Conclusions: While IHC has limitations, it offers a viable alternative for laboratories lacking access to advanced molecular techniques. Further research is required to determine the optimal combination of markers, establish a consensus diagnostic algorithm, and validate IHC through large-scale trials. This will ultimately enhance diagnostic accuracy, guide treatment decisions, and improve patient outcomes.

## 1. Introduction

Bladder cancer (BC) is the ninth most common malignancy worldwide, with approximately 614,298 new cases reported annually. It is the most frequently diagnosed tumour within the urinary tract, with particularly high incidence in Western Europe and North America, where the burden of the disease continues to grow [1].

BC is divided into two fundamental categories: non-muscle-invasive bladder cancer (NMIBC) and muscle-invasive bladder cancer (MIBC). NMIBC is the predominant phenotype, affecting about 75% of patients, and is typically characterised by frequent tumour recurrence after initial transurethral resection of the bladder tumour [2]. In cases with high-grade histology and/or early subepithelial invasion, intravesical immunotherapy with Bacillus Calmette–Guérin is often used.

The remaining 25% of BC corresponds to MIBC, where the standard treatment is radical cystectomy (RC). After RC, the five-year survival rate is around 50%. Recurrence is a significant concern, both locally (10–15% of patients) and systemically (20–50% of patients) [3]. Patients with MIBC that initially presented as NMIBC tend to have a poorer prognosis compared to those with primary MIBC, highlighting the importance of identifying NMIBC patients who do not respond to conventional treatment [4,5]. Neoadjuvant chemotherapy (NAC) followed by RC is recommended to improve outcomes, especially in advanced stages (T2-4N0M0) [6,7,8,9]. However, predicting which patients will benefit from NAC is challenging, leading to potential delays in curative surgery and poorer survival for those with residual tumours after treatment [10].

Despite advancements in clinical and pathological assessments, including histological studies based on tumour size, lymph node involvement, and the presence or absence of metastasis as outlined in the TNM classification, these factors have limited ability to accurately predict patient outcomes. The substantial heterogeneity of BC and the subjective nature of histological evaluations make it difficult to fully capture this diversity. This limitation has led to an exploration of genomic studies, which have revolutionised our understanding of BC by enabling its classification into distinct molecular subtypes. These classifications are essential not only for predicting patient prognosis but also for tailoring more effective therapeutic strategies. Additionally, genomics has helped in identifying biomarkers critical for early detection, risk stratification, and assessment of treatment response [11].

Several research groups have proposed different molecular classifications, suggesting various subgroups that have improved our understanding of BC biology and highlighted the clinical significance of these subgroups. They have demonstrated differing responses to chemotherapy and immunotherapy, supporting the need for a molecular classification of BC as a routine part of clinical practice. However, reproducibility issues, high costs, and limited availability of molecular analysis in diagnostic centres have restricted their use in everyday clinical practice [12,13].

The introduction of a consensus classification has sparked debate on how to implement this classification effectively [14]. One proposed solution is the use of IHC as a surrogate marker for molecular subtypes, particularly for the two main subtypes, basal and luminal, previously described in breast cancer [15]. It is already well established that IHC plays an important role in the diagnosis of bladder neoplasms, serving as a valuable complement to the morphological evaluation of the tumour alongside the patient’s clinical presentation [16]. In normal urothelium, cytokeratin (KRT) expression varies among cell types: basal cells express KRT14 and KRT5/6 but not KRT20, while terminally differentiated luminal cells express KRT20 but not KRT14 or KRT5/6. Intermediate cells variably express KRT5/6 without KRT14 or KRT20 expression [17].

Pioneering studies have demonstrated that basal and luminal profiles can be identified using IHC markers based on KRT, correlating with basal and luminal differentiation during urothelial carcinogenesis [18]. Specifically, KRT5/6+ and KRT20− expression is associated with basal subtypes, whereas KRT20+ and KRT5/6− expression correlates with luminal subtypes [19]. Meta-analyses have also confirmed that identifying basal and luminal subtypes using two immunohistochemical markers, GATA3 for luminal and KRT5/6 for basal, is highly accurate when compared with parallel gene expression studies [20].

Given the current lack of consensus and the need for further research to establish the true value and utility of these emerging taxonomies, continuing this line of inquiry is crucial. BC remains a condition with high morbidity and mortality, although significant progress has been made in recent years with the development of immunotherapies, such as immune checkpoint inhibitors and the introduction of antibody-drug conjugates (ADCs), aimed at improving patient outcomes [21]. In this context, immunohistochemistry (IHC) has emerged as a key tool, acting as a surrogate marker in the molecular subtyping of BC. Its ability to identify and classify tumours based on their molecular characteristics allows for a more precise evaluation of tumour behaviour and the selection of more targeted treatments. This represents a significant advance in precision cancer therapy and attempts to overcome some clinical limitations, such as reproducibility issues or the costs associated with these classifications [19,20].

In summary, while the molecular classification of BC has provided new insights into the disease’s biology, its implementation in clinical practice remains limited due to challenges such as the high cost of genomic sequencing technologies, the technical complexity of analysing molecular data, the need for specialised equipment and trained personnel, and the lack of standardised protocols across laboratories. The role of IHC has emerged as a promising tool to overcome these barriers by serving as a surrogate marker for identifying molecular subtypes of BC. This review will explore the use of IHC in the molecular classification of BC, assessing its accuracy as a surrogate marker in the molecular subtyping of BC, its clinical applicability, and its potential to enhance personalised treatment. By focusing on current evidence and ongoing challenges, we aim to determine whether IHC can be established as a standard practice in the molecular subtyping of this malignancy.

## 2. Materials and Methods

The review process was designed to ensure the accurate analysis and critical evaluation of information presented in the studies. A bibliographical search of articles was conducted in August 2024 using Medline and PubMed to retrieve studies published over the past eight years (2016–2024). The keywords used in the search included ‘bladder cancer,’ ‘molecular subtypes’, ‘immunohistochemistry’, and ‘molecular classification’. These terms, along with related ones, were employed to ensure comprehensive coverage of the topic and to identify studies relevant to the molecular subtyping and classification of BC.

In our narrative review, the inclusion criteria focused on original research articles, systematic reviews, meta-analyses, and clinical trials that addressed molecular subtyping and classification in BC, with a particular emphasis on those utilising IHC. Selected studies also needed to discuss the clinical applicability of these molecular subtypes in the diagnosis or treatment of BC, with a specific focus on IHC as a surrogate marker for molecular classification. We prioritised recent publications and studies relevant to the evolving molecular landscape of BC.

Exclusion criteria included the absence of immunohistochemical staining, studies not focused on BC, lack of full-text availability, and manuscripts not written in English.

Initially, our search for papers on BC and molecular subtypes yielded a total of 371 articles. After refining the search to include IHC, the number of relevant articles was reduced to 61. Among these, 5 were narrative reviews, 1 was a meta-analysis, and 1 was a systematic review (it was excluded as it did not meet our selection criteria).

After a thorough evaluation, we included the most recent and relevant articles that not only covered classical IHC markers for molecular subtyping but also highlighted the latest advances and the current status of their application in BC, with particular attention to their potential implementation in routine clinical practice (see Figure 1).

## 3. Cellular Origin, Carcinogenesis and Immunohistochemistry Correlation

The urothelium is composed of four to seven cellular layers. It features a single basal layer above the basement membrane, overlaid by two to five layers of intermediate cells and a superficial layer of umbrella cells. Each of these layers is characterised by different biomarker expressions [16].

Basal Layer: Expression of low-molecular-weight keratins such as KRT5/6, KRT14, anchoring molecules like laminin receptor or integrin beta4, and CD44 (hyaluronic acid receptor). Stem cells or uroprogenitor cells with self-renewal capacity reside in the basal layer. Intermediate Layer: These cells show moderate differentiation and express KRT18, with reduced levels of CD44 and KRT5/6. Superficial Layer: Differentiated cells express KRT20 and uroplakins [22].

Although the exact origin of BC remains uncertain, immunohistochemical and molecular analyses of the urothelial structure provide a foundation for studies on molecular subtyping (see Figure 2).

Urothelial carcinoma (UC) is the most common type of bladder cancer (BC). It develops via two distinct pathways: NMIBC and MIBC. NMIBC, which generally has a favourable prognosis, is molecularly and genetically stable and is associated with early genetic alterations such as deletion of chromosome 9 and point mutations in FGFR3, commonly found in papillary lesions. These changes suggest a clonal relationship between hyperplastic precursors and low-grade papillary carcinoma [23]. MIBC, which has an unfavourable prognosis, is genetically unstable and often involves the inactivation of key tumour suppressor genes such as TP53, RB1, and PTEN. Its development is typically preceded by flat urothelial dysplasia or carcinoma in situ (CIS), both of which share molecular features with invasive bladder carcinomas.

This dual pathway model of bladder carcinogenesis combines molecular, genetic, and pathological data and leads to the identification of two distinct phenotypes: NMIBC, which represents 75% of bladder cancer cases and usually appears as papillary tumours with limited potential for progression, and MIBC, which arises in 25% of initial diagnoses and is characterised by its aggressive nature and poor prognosis due to its propensity for invasion and metastasis [2].

In this context of carcinogenesis, IHC has emerged as a useful tool in diagnosis, particularly in CIS and metastatic UC. For the diagnosis of CIS, the basic panel of markers and their staining patterns, including full-thickness KRT20, loss of CD44, and aberrant p53, are used, although aberrant p53 expression is not a definitive marker for CIS. In the case of metastatic UC, markers such as KRT7, KRT20, GATA3, or p63 are employed. The cost-effective selection of biomarkers, tailored to differential diagnosis, facilitates accurate diagnosis and also contributes to the identification of different histological variants [16].

## 4. Molecular Classification

### 4.1. Genomic Era

The heterogeneity and complexity of BC are evident in the inconsistent prediction of treatment outcomes. Most efforts have focused on achieving significant advancements in understanding the genetic and molecular alterations involved in BC, particularly in studies on MIBC, providing a biological basis for characterising heterogeneity and identifying prognostic and therapeutic biomarkers through transcriptomic studies. Genomic research has enabled the classification of BC into different molecular subtypes with prognostic and therapeutic impact. Additionally, genomics has helped identify biomarkers that can aid in early detection, risk stratification, and treatment response.

Targeted therapies and immunotherapies have been developed based on specific genomic alterations found in each patient. Several research groups and scientific consortia have dedicated themselves to the study and development of molecular classifications in BC through gene expression analysis, each defining different subtypes [22].

So far, validation studies of molecular classifications in BC are considered limited, but some ongoing trials incorporate molecular classification and are expected to provide information on its applicability in clinical practice. Given the lack of consensus and the need for further research to establish the true value and utility of these emerging taxonomies, it is crucial to continue this line of research, as BC remains a pathology with high morbidity and mortality. Despite significant progress made in recent years with the development of immunotherapies and antibody-drug conjugates, further advancements are still needed to improve patient outcomes and address treatment limitations [21].

### 4.2. Development of Molecular Classifications

The initial efforts to classify BC began in 2003 with microarray analysis at a time when there were no established IHC or molecular markers to define clinically relevant BC subsets. The primary goal was to analyse the molecular characteristics of bladder tumours and identify genetic patterns to better understand and classify tumour heterogeneity. Researchers used microarrays to simultaneously examine the expression of thousands of genes in tissue samples from bladder carcinoma. This approach revealed distinct bladder carcinoma classes with characteristic gene expression profiles, reflecting differences in tumour biology and disease progression. Correlations between certain gene expression profiles and patient survival suggested that molecular classification could have prognostic implications. Specific genes were also identified as being overexpressed or underexpressed in different bladder carcinoma classes, offering insights into the molecular pathways and biological processes involved. This pioneering study laid the groundwork for future research in molecular classification and personalised treatment of BC [24].

In 2005, a gene expression microarray study classified BC into more homogeneous subgroups, distinguishing between superficial (Ta, T1) and muscle-invasive tumours (T2–T4). These subtypes, which included transitional versus squamous and good versus poor prognosis groups, were linked to different clinical outcomes and survival rates [25].

These early studies laid the foundation for BC taxonomy, encouraging further research by major working groups that established potential new molecular taxonomies. Subsequent research has focused on integrating these molecular classifications with IHC to better stratify BC patients, enhancing the role of IHC as a surrogate marker in the diagnosis and treatment of BC. This integration has facilitated the transition from purely genomic approaches to those that combine molecular, genetic, and histologic data, ultimately leading to more precise and personalised treatment strategies for patients. The work of groups such as The Cancer Genome Atlas (TCGA) has been instrumental in defining these molecular subtypes.

Studies conducted by the Baylor College of Medicine [26], the University of North Carolina [15], MD Anderson [19], and the Lund group [27,28] have identified subgroups based on urothelial cell differentiation and specific genomic characteristics. These classifications have revealed significant differences in the expression of IHC markers in the two main subtypes, basal (KRT5/6, CD44+, RB) and luminal (PPARG, GATA3, KRT20, FGFR3, TSC1), which have clinical implications in terms of treatment response and prognosis. The basal subtype, with high expression of basal KRTs, has a worse prognosis and a greater similarity to basal-like subtypes in breast cancer, while the luminal subtype has a better prognosis and is associated with *FGFR3* mutations.

In 2014, TCGA [29] published a classification that categorised BC patients into four groups based on their molecular alterations. Cluster I, with luminal characteristics, papillary histology, *FGFR3* mutations, and *CDKN2A* deletions, and Cluster II, also luminal but less differentiated, with *p53* alterations and elevated *HER2*, showed similarities to the luminal A subtype of breast cancer. Cluster III, termed basal/squamous, was characterised by squamous histology and KRT5/KRT14 expression, similar to the basal-like profile in breast cancer. Cluster IV included poorly differentiated tumours with molecular alterations in immune system genes and epithelial-mesenchymal transition genes. The study suggested MIBC could be treated using strategies similar to those for breast cancer. Additionally, a subtype of urothelial carcinoma with cancer stem cell characteristics was identified in Cluster III, suggesting another potential therapeutic pathway. This analysis has improved the understanding of the molecular heterogeneity in BC, highlighting the high genomic alteration burden and laying the groundwork for future research and more personalised therapeutic approaches [29].

In the 2015 consensus meeting organised at the Spanish National Cancer Research Center, leading research groups, including TCGA, Lund University, Baylor College of Medicine, MD Anderson Cancer Center, University of North Carolina, and Institut Curie, agreed on IHC criteria for identifying the Basal Squamous subtype in bladder cancer. The consensus defined IHC-based criteria with positive expression of KRT5/6 and KRT14 and absence of FOXA1 and GATA3, establishing IHC as a surrogate marker for molecular subtyping. This approach facilitates reliable identification of this subtype in clinical practice, especially in settings lacking access to advanced molecular techniques [30].

In 2017, the second TCGA classification significantly expanded the understanding of molecular subtypes by analysing 412 cases, identifying 34 additional tumour suppressor genes and 158 genes subject to epigenetic silencing. This classification redefined the previously established molecular subtypes, incorporating both messenger RNA and long non-coding RNA profiles. Tumours were grouped into luminal subtypes (luminal-papillary, luminal-infiltrated, and luminal) and basal subtypes (basal/squamous and neuronal), each with distinct molecular characteristics and prognoses [31]. The neuronal and luminal subtypes were corroborated by the independent cohort from the second Lund classification that same year [28]. The Lund group employed a global mRNA expression analysis for phenotypic tumour profiling and to define molecular subtypes in nearly all major tumour types. They analysed 307 patients with advanced BC who underwent cystectomy, using gene expression analysis and IHC detection with antibodies for 28 proteins. They described five tumour cell phenotypes in advanced UC: Urothelial-like (URO), Genomically unstable (GU), basal-squamous (Basal/SCC-like), Mesenchymal-like (Mes-like), and Small-cell/neuroendocrine-like (Sc/NE-like).

This second TCGA classification expanded the study and understanding of the different molecular subtypes by examining the high mutation rate in BC, facilitating a therapeutic approach tailored to each subtype. It also established an IHC concordance with the molecular clustering of the different subtypes. Luminal tumours were identified as KRT20+/GATA3+/FOXA1+ and basal tumours as KRT5/6+/KRT14+/GATA3−/FOXA1− [31].

### 4.3. Taxonomic Consensus

The diversity of published classifications has hindered the implementation of molecular classification as a routine practice in the clinical setting. Most working groups have presented similar and complementary molecular findings, but the main differences have been related to the representation of NMIBC and MIBC tumours. These differences are largely due to issues in case selection, variable heterogeneity in tissue preservation, and quality. Another important aspect regarding molecular subtypes is that most of the available data in the different models have been generated from repositories containing frozen bladder tumours, with minimal involvement of pathologists in selecting appropriate samples for molecular analysis, as well as the use of different biomarkers for basal and luminal profiles in the various classifications [32].

Therefore, a consensus classification was necessary to unify the different classifications previously published. In 2020, Kamoun et al. [14] published a consensus classification to achieve an international consensus on the molecular subtypes of MIBC that reconciles the existing classification schemes. The Bladder Cancer Molecular Taxonomy Group developed a classification system using transcriptomes from 1750 patients and a network-based analysis of multiple independent classification systems based on six molecular classifications: Baylor [26], the University of North Carolina [15], MD Anderson [19], Robertson et al. from TCGA [31], Lund [28], and the Institut Curie [33]. The consensus groups are: Luminal Papillary (LumP): characterised by high expression of FGFR3-related signatures, enriched in *FGFR3* mutations and papillary morphology; Luminal Non-specified (LumNS): displayed elevated stromal infiltration, particularly fibroblastic, and was associated with micropapillary histology and CIS; Luminal Unstable (LumU): marked by high cell cycle activity, frequent *PPARG* alterations, *ERBB2* amplifications, and *TP53* mutations; Stroma-rich: characterised by significant stromal infiltration with smooth muscle, endothelial cells, and fibroblasts, showing intermediate urothelial differentiation; Basal/Squamous (Ba/Sq): enriched in *TP53* and *RB1* mutations, associated with squamous differentiation, and showed high immune infiltration, particularly cytotoxic lymphocytes and NK cells; and Neuroendocrine-like (NE-like): linked to neuroendocrine differentiation, with TP53 and RB1 co-alterations, and associated with poor prognosis.

The consensus classification provides a robust framework for testing and validating predictive biomarkers, potentially improving personalised treatment strategies. However, the study’s focus on correlating molecular classifications with histological features underscores the importance of IHC in validating and applying molecular classifications in clinical practice (see Table 1).

In 2024, López-Beltrán et al. [23] presented a comprehensive analysis of diagnostic and treatment advances in bladder cancer, detailing the main molecular classifications, their subtypes, and associated IHC markers. This approach provides a clear view of the relationship between molecular subtypes, diagnostic characteristics, and therapeutic strategies, providing a solid basis for clinical application (see Table 1).

### 4.4. New Molecular Classifications: The Path Forward

So far, molecular classifications are generally based on transcriptomic profiles, generating very diverse categories with limited correlation. Despite the publication of the Consensus Molecular Classification, the application of molecular classification still requires expensive and highly complex technology, which is often not available in routine clinical practice. Despite the distinction between two fundamental subtypes, basal and luminal, many subtyping systems include peculiarities, such as subtypes not recognised by any other system. There are broad and blurred boundaries between molecular subtypes, and cases that fall within these limits are often classified differently by different subtyping systems. Additionally, when there are histomorphologically distinct regions within a single tumour, the molecular subtypes of these regions are often discordant. These factors highlight the biological diversity of BC, the association of molecular subtypes with tumour stage and histomorphology, and the lack of consensus in some aspects among subtyping systems [12,34].

The introduction of the NanoString technology, which has been established for nearly two decades, offers an alternative for molecular subtyping in BC. Multiple studies, including López Beltrán et al. [35], have employed the NanoString technology with direct digital detection chemistry to conduct a study with a retrospective cohort of 91 patients, including MIBC (T2-T3) and NMIBC (Ta-T1), using mRNA detection of four markers incorporating GATA3 and KRT20 (typically related to the luminal molecular subtype) and KRT5 and KRT14 (typically related to the basal molecular subtype) [35]. This technology determined molecular subtypes according to mRNA expression of GATA3 or KRT20 for luminal subtypes (71%) and KRT5 or KRT14 for basal subtypes (21%).

The NanoString technology showed promise in accurately classifying molecular subtypes of urothelial carcinoma of the bladder quickly and in a reproducible and robust manner. It also presented an adequate correlation with immunohistochemical markers, with luminal subtypes being GATA3+, KRT20+, basal subtypes KRT14+, KRT5+, GATA3 low/−, KRT20 low/−, and null/double-negative (DN) being GATA3−, KRT20−, KRT14−, KRT5− [35].

The study by Olkhov-Mitsel E et al. [36] explores the use of the NanoString nCounter platform, while conventional IHC with GATA3 and KRT5/6 was re-evaluated for the molecular classification of MIBC in clinical practice settings. A molecular classifier was developed based on NanoString technology and seven genes (KRT5, KRT6, SERPINB13, UPK1A, UPK2, UPK3A, and KRT20) using a cohort of 138 MIBC cases that allowed the assignment of two molecular subtypes, luminal or basal, each with different prognoses and therapeutic responses. Furthermore, the previously published IHC studies of KRT5/6 and GATA3 [37,38] were validated, demonstrating a high concordance of 96.9% with the molecular classification by gene expression based on NanoString technology.

Despite the limitations of studies using NanoString-based gene expression, such as their retrospective nature and relatively small sample size, the results support the clinical applicability of NanoString technology in the classification of molecular subtypes in BC, providing a high correlation with immunohistochemical studies and laying the groundwork for future clinical applications of BC molecular classification [39] (see Figure 3).

## 5. Immunohistochemistry as a Surrogate Marker

### 5.1. Immunohistochemistry as a Practical Tool for Subtyping

The lack of reproducibility, along with the high cost and limited availability of molecular analyses in routine diagnostic centres, restricts the use of molecular classifications for BC in routine clinical practice [12,20].

Conventional pathological parameters, such as tumour stage and grade, have limited ability to predict the heterogeneous behaviour of BC, especially for tumours with similar stages and grades [23]. In this context, IHC analysis of biomarker expression complements the morphological evaluation of the tumour, offering information to support the accurate diagnosis of BC [16].

Growing evidence suggests that IHC markers are not only useful for diagnosis but also for identifying patients at high risk of post-surgical progression, thereby improving disease management in various cancer types, such as breast cancer [40].

Following the proposal of the Consensus Molecular Classification, the debate emerged on how to implement this classification in clinical practice. One proposed approach is the use of IHC as a surrogate marker for molecular subtypes, particularly for the two main subtypes already described in breast cancer: basal and luminal. It is well-known that IHC plays a crucial role in the diagnosis of bladder neoplasms, complementing the morphological evaluation of the tumour and the patient’s clinical profile [12] (see Table 2).

### 5.2. Immunohistochemistry-Based Classification in MIBC

Most studies are based on transcriptomic analyses of MIBC patients, which aim to categorise these two major molecular groups, known for their distinct clinical behaviours and varying sensitivities to chemotherapy. Several studies have attempted to validate these subtypes in various clinical cohorts and identify characteristic IHC markers to simplify and optimise this categorisation. A comprehensive meta-analysis utilised only two IHC markers, luminal (GATA3) and basal (KRT5/6), which were sufficient to identify molecular subtypes with over 90% accuracy [20]. Luminal tumours exhibited an expression pattern similar to the intermediate/superficial layers of the urothelium, frequently activating *PPAR* target genes and showing a higher incidence of mutations in *FGFR3*, *ELF3*, *CDKN1A*, and *TSC1*, as well as overexpression of proteins such as E-Cad, HER2/3. Conversely, basal tumours displayed an expression signature akin to the basal layer of the urothelium, with mutations in *TP53* and *RB1* and overexpression of CD49, Cyclin B1, and EGFR. Survival analyses revealed that muscle-invasive basal BC exhibited more aggressive behaviour compared to luminal BC. This study implemented molecular classification based on mRNA transcriptome analysis alongside IHC analysis to corroborate the molecular classification.

In the study by Guo et al., the two main subtypes of BC were again identified: basal and luminal. They focused on GATA3 as a marker for the luminal subtype and KRT5/6 for the basal subtype. Additionally, they conducted a genetic analysis using a panel of 28 luminal markers and 20 basal markers. They developed a quantitative algorithm called basal to luminal transition based on mRNA expression to identify these molecular subtypes. This algorithm was further validated with IHC, facilitating its use in clinical practice [35]. Prior to this study, Hodgson et al. used the same antibodies, identifying 85.2% of tumours as luminal (KRT5/6−, GATA3+) and 14.8% as basal (KRT5/6+, GATA3−); curiously, the basal subtype showed better disease-specific survival, higher CD8+ lymphocyte counts, and increased PD-1 and PD-L1 expression [41].

Choi et al. was one of the pioneering studies to demonstrate basal and luminal profiles using IHC with two markers, KRT5/6 and KRT20. On the one hand, KRT5/6+ and KRT20− detected most of the basal subtypes, while on the other hand, KRT20+ and KRT5/6− categorised the luminal subtype [19]. This study concluded that basal tumours with KRT5/6, p63, and KRT14 expression had a better response to chemotherapy, while luminal tumours with active PPAR expression and activating mutations in FGFR3 showed expression of KRT20, FOXA1, and GATA3.

The Lund working group has been one of the most significant in leading not only the taxonomic identification of molecular subtypes but also in implementing molecular subtyping based on IHC along with global mRNA expression analysis. Initially, they used 28 antibodies, including CCNB1, CCND1, CDH1, CDH3, CDKN2A, CHGA, FGFR3, FOXA1, GATA3, KRT5, KRT14, KRT20, SYP, and TP63, to define molecular subtypes. They observed discrepancies between IHC analysis of the tumour cell phenotype and subtypes defined by mRNA expression [28]. This same group explored the relationship of molecular subtypes in MIBC with lymph node metastasis, using 29 markers and finding high concordance in luminal subtypes between primary tumours and metastases. However, significant discordance in the Ba/Sq subtype suggests caution when using IHC for therapeutic planning in lymph node metastatic MIBC [42].

Moreover, they created an IHC-based subtyping algorithm system that initially groups three categories based on GATA3 and KRT5: URO and/or GU (GATA3+/KRT5 low), Ba/Sq (GATA3−/KRT5 high), and Mes-like and/or Sc/NE-like (GATA3−/KRT5 low). At a second level, p16 can be applied to distinguish between urothelial (p16−) and genomically unstable (p16+), and EpCAM between mesenchymal (EpCAM−) and NE (EpCAM+). At a third level, optional IHC can be applied in URO (CCND1+), GU (CCND1−), Ba/Sq (KRT14+), Mes-like (vimentine+, and sarcomatoid morphology), and Sc/NE-like (NE markers and small cell morphology) [43].

In recent years, due to the complexity of IHC algorithms and occasional lack of concordance with genetic analysis, there has been an updated review of the molecular classification of BC, emphasising the importance of already recognised subtypes and discussing additional classifications into subtypes such as URO and GU based on IHC patterns. Efforts have been made to optimise, validate, and update these algorithms. It has been proposed that an algorithm based on GATA3, KRT5, and p16 can achieve at least 78% accuracy in identifying the URO, GU, and Ba/Sq subtypes of the Lund classification and 91% accuracy for the luminal/basal classification, providing not only the identification of luminal and basal subtypes but also the determination of several subcategories, allowing for more precise risk stratification [44].

Bernardo C et al. published two studies that enhance the Lund taxonomy using an extensive IHC study to complement and correct RNA-based classifications. One of these studies focuses on luminal subtypes, particularly URO (URO A, URO B, URO C) and GU, with a cohort of 344 luminal tumours using 17 IHC markers, most of which are not routinely used in clinical practice. URO exists within a spectrum of heterogeneity that may explain why different researchers have identified different numbers of luminal subtypes. URO tumours show biological progression from URO A towards URO Ap and URO C, while GU represents a biologically distinct subtype [45]. The other study characterised non-luminal subtypes Ba/Sq and Sc/Ne with a cohort of 347 tumours using 15 IHC markers, also not typically used in clinical practice. They highlight that while Ba/Sq tumours exhibited basal characteristics, they lack urothelial differentiation and show high keratinisation and disorganised proliferation, whereas Sc/Ne tumours represent a more extreme version of GU tumours, with rampant proliferation and an almost complete lack of urothelial differentiation [46]. These findings underscore the importance of IHC in the classification of urothelial tumours and highlight the complexity of BC. Both cohorts are derived from previous Lund studies [27,28].

The evolution of the Lund taxonomy has been aimed at harmonising molecular classification based on gene expression and IHC, resulting in the creation of a classification system called LundTax. This system combines gene expression analysis with the integration of IHC-based phenotypes, excluding non-cancerous factors such as immune infiltration and cell proliferation, which often confuse other classifications for both NMIBC and MIBC [47].

The IHC-based subtyping, according to the Lund classification, has been used by numerous research groups. Hesswani et al. investigated the feasibility of an IHC-based subtyping method using a three-antibody algorithm (KRT5, GATA3, and p16) with external validation and examined the correlation of subtypes with radiation therapy (RT) outcomes. The accuracy was 89% in molecular subtypes (23.6% basal, 14% GU, 31.2% URO, 31.2% non-classified), although no significant association was found between the subtypes treated with RT and survival [48]. This cost-effective 3-IHC-based algorithm has been employed in other recent studies, such as that by Telrevic R. et al., which uses GATA3 and KRT5/6 to distinguish between luminal and basal types and incorporates p16 as an economical way of further subclassifying the luminal type into LumP and LumU types, consistent with data from the University of Lund classification [49]. In a study by Olkhov-Mitsel et al., the authors used the same three-antibody classifier on tissue microarray material from chemotherapy-naive cystectomy specimens. In their series, 97.1% of MIBC cases were classified as either luminal GU (GATA3+/KRT5/6−/P16+), luminal URO (GATA3+/KRT5/6−/P16−), or basal (GATA3−/KRT5/6+) according to the Lund scheme in a cohort of 243 MIBC [50].

Queipo et al. conducted another study aimed at developing and validating an IHC-based algorithm using KRT5/6, KRT14, GATA3, and p16 for subtyping. URO is characterised by high positivity for GATA3 and low for KRT5/6 and p16; GU by high positivity for GATA3 and p16 but low for KRT5/6 and KRT14; Ba/Sq by high and diffuse positivity for KRT5/6 and KRT14, with lower expression of GATA3 and p16; and Mes-like and Ne-like by the lack of expression of GATA3, KRT5/6, and KRT14. IHC-based subtyping classified 72.57% as luminal and 27.43% as non-luminal. This group also emphasised the crucial role of the pathologist in interpreting KRT5/6 staining patterns, as diffuse staining in the Ba/Sq subtype and parabasal or patchy patterns might indicate a luminal pattern depending on the co-expression of other markers like GATA3 [51].

IHC has also been used as a surrogate marker in the study of BC in patients under 45 years old, following a panel of 23 IHC markers used by the Lund group. This further reinforces its applicability in IHC panels, even without validation from genomic analyses [52].

Previously published IHC classification schemes have employed numerous antibodies, many of which are not commonly used in routine clinical practice. This has led other research groups to simplify the IHC panel and focus on categorisation into the two main subgroups [38,53,54,55,56,57,58,59,60,61,62,63]. Simplified IHC panels typically include at least two antibodies: one for the luminal profile, often GATA3, and another for the basal profile, such as KRT5. These simplified panels have been widely used, either as a primary objective for validation or as a secondary goal. The use of these panels has led to the identification of subtypes, such as DN and double-positive (DP) tumours, which are defined by the absence or coexistence of typical luminal and basal markers, respectively. Many of these tumours show morphological features of urothelial differentiation [37,38,55,56,57,58,60]. It has been proposed that DN tumours might be included in the NE-like consensus class. Another possibility is that both DP and DN tumours are distinct subtypes of the luminal and Ba/Sq subtypes, identified by a reduced number of markers, or that these tumours are in transition between different molecular subtypes [64,65].

Sangredolce et al. reported that DP and DN tumours represented 26.9% and 3.2% of cases, respectively. All DN cases and 83% of DP cases exhibited a non-invasive pattern of invasion, but no significant differences in overall survival (OS) were observed between these subtypes [55]. In contrast, Bejranada et al. found that the DN subtype accounted for 10% of cases and was associated with worse 5-year OS [56]. Jangir et al. presented a cohort of 40 patients, where DP tumours represented 15% of cases and demonstrated intermediate survival, while DN tumours were rare (2.5%) and not associated with a specific prognosis due to their low prevalence [57]. Serag et al. analysed a cohort of 80 NMIBC and MIBC patients using a 2-IHC-based algorithm, finding that luminal (GATA3+/KRT5/6−) and basal (GATA3−/KRT5/6+) subtypes represented 60% and 7.5% of cases, respectively. Additionally, 25% and 7.5% of cases corresponded to DP and DN subtypes, respectively. The authors suggested that the presence of these latter two groups might be due to the lack of clear cutoff criteria in their study [66]. They also noted a trend toward better survival in the luminal subtype compared to other groups, which is consistent with findings from other studies [67]. Overall, DN and DP tumours appear to have heterogeneous features in terms of prevalence, clinical behaviour, and survival, highlighting the need for further validation studies and better characterisation.

The relationship between molecular subtypes and their prognostic implications has been described in multiple studies [19,20,68,69]. Font et al. identified MIBC patients who might benefit from NAC using IHC-based classification into three subgroups: Ba/Sq type (FOXA1 and GATA3 low, KRT5/6 and KRT14 high), luminal type (FOXA1 and GATA3 high, KRT5/6 and KRT14 low), and a mixed group (FOXA1 and GATA3 high, KRT5/6 high, and KRT14 low). They indicated that patients with Ba/Sq-type tumours were more likely to achieve a complete pathological response to NAC [70]. The proposed classification was supported by another study that used FOXA1 and KRT14 as markers for luminal and basal subtypes in patients with MIBC and conventional UC variants. They found that high KRT14 and low FOXA1 were expressed in basal subtypes, while luminal tumours had low KRT14 and high FOXA1, categorising micropapillary, nested, and plasmacytoid carcinoma as luminal [71].

This better response to NAC by basal tumours has also been supported by other studies that used IHC as surrogate markers. Helal DS et al. proposed an IHC algorithm using GATA3, KRT5, and p53 in patients, stratifying them into three subtypes: luminal with GATA3 expression, p53-WT subtype showing weak nuclear staining in less than 50% of tumour cells as a surrogate for a non-mutated TP53, and the basal subtype, which showed a significantly better response to NAC, while the p53-WT subtype was chemoresistant. They also studied the IHC relationship of HER2, which was significantly more expressed in luminal subtypes [72]. Koll JF et al. tested the predictive value of the two-marker-based IHC classification (GATA3, KRT5/6) for chemotherapy efficacy in the cohort of adjuvant-treated patients. Although patients receiving adjuvant chemotherapy had a significant survival benefit, a two-classification system might not sufficiently reflect the heterogeneity of BC to make treatment decisions [38].

Additionally, the use of IHC has helped to improve the understanding of the classification and differences between primary tumours and their metastases, especially in the Ba/Sq subtype [42,73]. Bontoux et al. distinguished between luminal and basal/squamous subtypes using an IHC panel. Luminal tumours were (GATA3, FOXA1+), and basal/squamous tumours were (KRT5/6, KRT14+). High concordance was observed between primary tumours and lymph node metastases using an IHC panel for subtype classification, with high concordance for KRT14 in the Ba/Sq subtype [74]. In contrast, Sjödahl et al. combined IHC and mRNA expression profiling and observed discordance specifically in Ba/Sq tumours, attributing this to phenotypic adaptation and intratumoural heterogeneity in lymph nodes, factors that may not be detectable with IHC alone [42].

### 5.3. Immunohistochemistry-Based Classification in NMIBC

The tumour heterogeneity observed in BC presents significant challenges in grading and staging, particularly in NMIBC, as its molecular characterisation remains underexplored, with most research focused on MIBC [75].

Approximately 75% of BC patients are diagnosed with NMIBC, a condition marked by heterogeneity where a practical dual IHC-based classifier for luminal and basal phenotypes has not proven successful in predicting outcomes [76]. Further studies are needed to determine whether these molecular subtypes can be effectively classified using IHC and if they possess prognostic value in NMIBC.

MD Anderson’s group has been a leading institution in the molecular classification of BC, demonstrating that basal and luminal subtypes can be identified through IHC using KRT5/6 and KRT20 markers. Efforts were made to validate these findings [19] on the use of KRT20 and KRT5/6 for IHC identification of luminal and basal subtypes, respectively, leading to the development of a molecular grading classification for NMIBC [77]. The results were consistent with another study that identified high mRNA expression of KRT20 and low expression of KRT5 in luminal subtypes [78], where the luminal phenotype was associated with worse progression-free survival and recurrence-free survival. The study concluded that luminal-basal categorisation by IHC is valid, and RT-PCR allows for a more refined classification. Additionally, it identified two more categories based on IHC and RT-PCR: the DP category (KRT20+, KRT5/6+) and the DN category (KRT20−, KRT5/6−). Breyer et al. demonstrated that RT-PCR-based categorisation is superior to using only IHC, offering greater accuracy for prognosis in NMIBC [78]. However, the relevance and classification of these categories are still unclear and require further research for better understanding [77].

Rodríguez Pena C. et al. conducted another study that contributed additional findings on IHC classification and prediction in NMIBC. They evaluated luminal markers (GATA3, KRT10, ER, Uroplakin2, and HER2) and basal markers (KRT5/6, CD44) in NMIBC. Molecular subtyping was a secondary objective, with the primary focus on correlating these markers with recurrence, progression, and stage. They highlighted that the progression associated with KRT5/6 could be significant for assessing its role as a prognostic marker and CD44 as a recurrence marker, although the cohort used was heterogeneous in terms of tumour stage and grade [79].

Following the simplified three IHC-based panels by Lund Taxonomy, an algorithm using three antibodies (GATA3, KRT5, and p16) was employed in NMIBC patients. This algorithm classifies patients into four subtypes: Basal, GU, URO, and URO-KRT5+, aiding in the stratification of subtypes into low or high risk of recurrence and progression. The study demonstrated that basal subtypes had the worst prognosis with faster progression, supporting the use of the IHC algorithm in routine clinical practice and therapeutic decision-making [80].

In the study by Muilwijk et al., a cohort of 109 pTa NMIBCs was evaluated. Although the study did not explicitly classify molecular subtypes, it focused on the application of IHC markers, including basal markers (p63, KRT5, p40) and luminal markers (GATA3, KRT20). They identified an inverse correlation between KRT5 and KRT20, which effectively distinguished high-grade from low-grade NMIBC, indicating distinct biological behaviours in NMIBC compared to MIBC. The expression of KRT5 was validated by mRNA expression analysis [81]. Studies on non-muscle-invasive upper urinary tract urothelial carcinoma show that KRT5 is an independent prognostic factor in this setting [82,83]. Sikic et al. concluded that IHC subtyping using KRT5 and KRT20 can identify upper urinary tract urothelial carcinoma subtypes with significantly worse prognosis, particularly KRT20+/KRT5− tumours [82]. Jung et al. also concluded that the expression of KRT5/6 and KRT20 is crucial for prognostic stratification, with basal profile tumours (KRT5/6 positive) showing better survival compared to luminal profile tumours (KRT20 positive) [83]. However, these results regarding KRT5 are controversial. Mai et al., in their study, evaluated KRT and CD44 in a cohort of 302 NMIBC cases and found that a basal-like subgroup, identified by strong reactivity to KRT5 and CD44, constituted 12.9% of low-grade tumours and 17% of high-grade tumours. This basal subgroup showed a notable tendency towards multifocality, higher recurrence, and accelerated progression to more advanced grades and stages compared to non-basal tumours [84]. Breyer J et al. focused on the expression of KRT20 (luminal) and KRT5 (basal) mRNA to subtype NMIBC, with IHC also used, showing that high KRT20 and low KRT5 levels were associated with worse prognosis [78], corroborating others studies [76,77,85]. The study concluded that RT-PCR provided superior prognostic accuracy compared to only IHC, suggesting that KRT5 in NMIBC can correlate both positively and negatively across different NMIBC groups [75,86].

There are few studies that subtype CIS. Garczyk et al. focused on subtyping CIS using a panel of luminal markers (KRT20, GATA3, ERBB2) and basal markers (KRT5/6, KRT14), along with p53 to assess typical alterations in CIS. They highlighted the intratumoural heterogeneity in CIS but did not find a direct correlation between molecular subtypes and clinical prognosis [87]. Similarly, Barth et al. employed a similar IHC panel, including luminal markers (KRT20, GATA3, ERβ, HER2) and basal markers (KRT5/6, KRT14). Their study stood out by analysing the phenotypic transition of CIS during its progression to invasive tumours, observing a significant loss of luminal markers and an increase in basal markers in invasive areas. This suggests a shift from a luminal to a basal phenotype, which could have therapeutic implications [88].

Molecular classification based on IHC can be a key predictor of the efficacy of intravesical chemotherapy in NMIBC patients, particularly in basal subtypes. Specifically, tumours classified as basal according to IHC markers have shown greater sensitivity and better progression-free survival when treated with gemcitabine and cisplatin compared to other molecular subtypes [89]. This molecular classification not only predicts treatment response but also underscores the importance of tailoring therapeutic strategies according to the molecular subtype, with gemcitabine showing greater effectiveness in basal subtypes compared to anthracyclines [90]. This correlation highlights the importance of molecular stratification through IHC to optimise therapeutic strategies and improve clinical outcomes.

IHC has been used in studies such as Ottley et al. to evaluate luminal, basal, and epithelial-mesenchymal transition markers in high-grade T1 cases, exploring their role as predictors of disease-specific survival and progression to MIBC. Luminal markers FOXA1 and SCUBE2 were significantly associated with better disease-specific survival, while markers related to the epithelial-mesenchymal transition process did not prove to be significant predictors of disease-specific survival [91].

Discrepancies in the literature regarding the use of IHC as a surrogate marker in NMIBC are partly due to the diversity in methods used (IHC versus RT-qPCR), tumour stage (Ta, T1, CIS), and tumour grade (high/low grade) in many of these studies (see Figure 4 and Appendix A).

## 6. Conclusions

Molecular classifications have revolutionised the approach to BC by providing crucial biological knowledge for developing personalised treatment strategies. However, most of these classifications are based on transcriptomic studies that, while valuable, are financially costly and not widely accessible in routine clinical practice. There is a need for a more accessible and readily implementable tool, such as the IHC-based algorithm, to harness the prognostic and predictive potential of molecular subtypes, particularly in patients undergoing neoadjuvant therapy.

Despite the promise of IHC-based approaches, many studies face significant challenges. The lack of external validation, small sample sizes, and non-standardised sample collection present considerable limitations. Additionally, the use of different markers, particularly in the two main subtypes, complicates the process. The retrospective nature of many studies, variations in sampling techniques (whether using full tissue block sections or tissue microarrays), and intratumoural heterogeneity can also affect the effectiveness of staining algorithms. These factors, combined with differences in the number and type of antibodies used, lead to variability in the accuracy of findings across different studies.

The challenge lies in determining the optimal combination and appropriate number of markers to effectively classify patients into various subtypes. It is essential to conduct well-designed comparative trials with large cohorts to validate IHC markers and establish a standardised diagnostic algorithm for both MIBC and NMIBC.

## Figures and Tables

**Figure 1 diagnostics-14-02501-f001:**
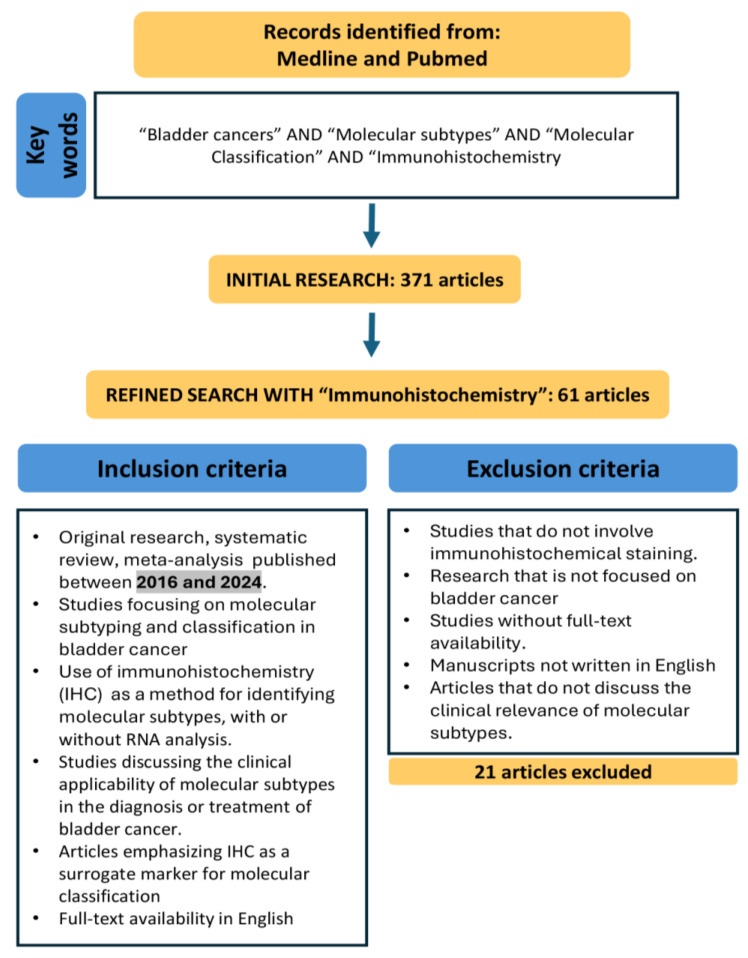
Flowchart of the narrative review process for identifying relevant studies on the molecular subtyping and classification of bladder cancer (BC) using immunohistochemistry (IHC) between 2016 and 2024.

**Figure 2 diagnostics-14-02501-f002:**
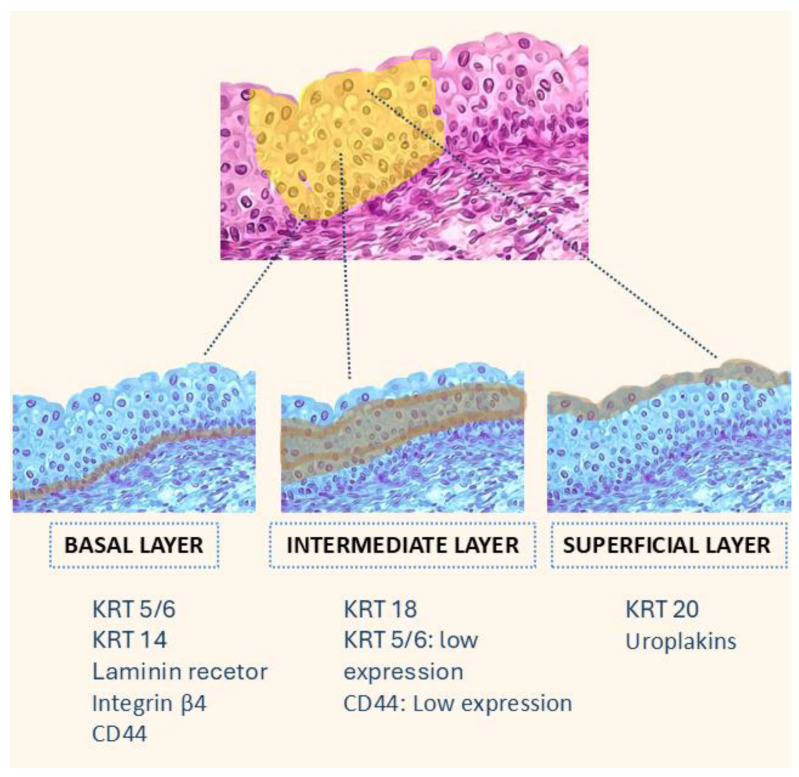
Cellular composition of the urothelium and associated keratin expression. The figure illustrates the three main layers of the urothelium: basal, intermediate, and superficial. Each layer is characterised by the expression of specific keratins and other markers. The basal layer expresses KRT 5/6, KRT 14, integrin β4, and CD44; the intermediate layer expresses KRT 18 with low levels of KRT 5/6 and CD44; and the superficial layer is marked by KRT 20 and uroplakins, which are associated with terminal urothelial differentiation.

**Figure 3 diagnostics-14-02501-f003:**
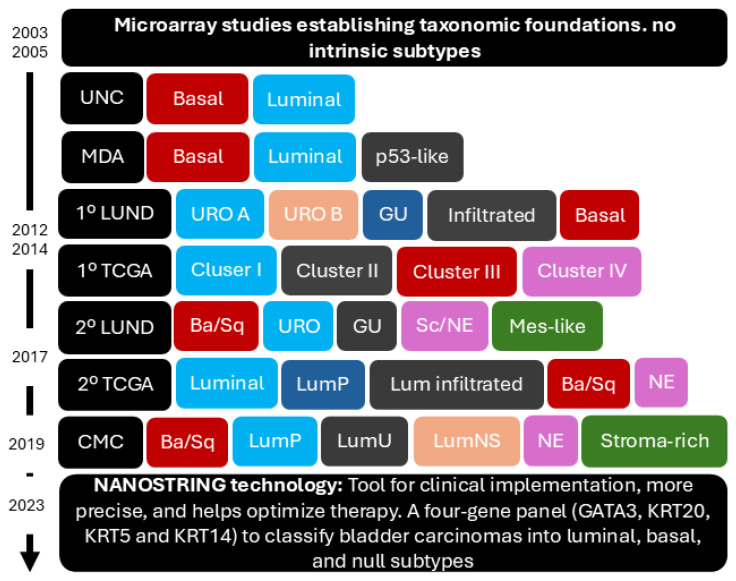
Evolution of molecular taxonomy in bladder cancer. This figure shows the progression from early microarray studies, which did not identify intrinsic subtypes, to the more sophisticated molecular classifications developed over time by institutions such as the University of North Carolina (UNC), MD Anderson Cancer Center (MDA), The Cancer Genome Atlas (TCGA), and (Lund University) Lund, culminating in the consensus classification (CMC). The timeline highlights the expansion from basal and luminal subtypes to more refined categories. The adoption of NanoString technology in 2023 marks a key advancement for clinical implementation, providing a more precise and efficient approach to molecular subtyping and therapy optimisation. The color coding within the figure reflects the categorization of molecular subtypes across studies: the red tones represent basal-like subtypes, which display basal characteristics across different classification systems; the blue tones correspond to luminal subtypes, associated with urothelial differentiation; gray indicates urothelial-like or p53-like subtypes, depending on the classification; green represents stromal-rich or infiltrated subtypes, which are distinguished by immune or stromal signatures; and orange and pink denote neuroendocrine and mesenchymal-like categories, introduced in more recent classifications.

**Figure 4 diagnostics-14-02501-f004:**
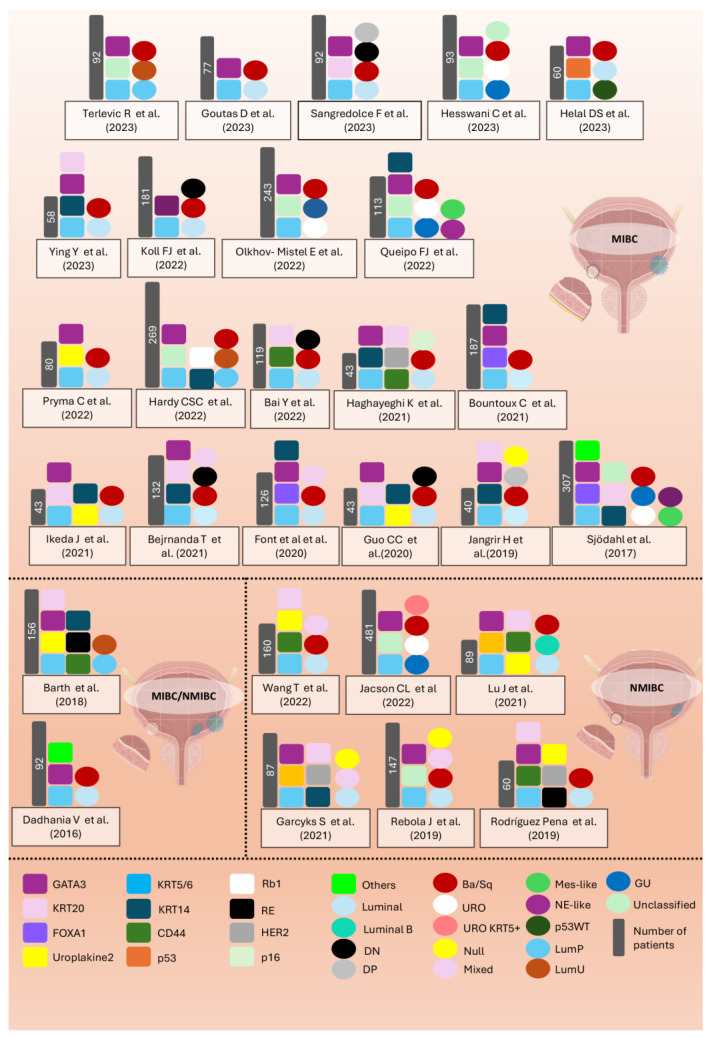
Overview of the main studies on molecular subtyping of bladder cancer (BC) using immunohistochemical (IHC) markers. The figure highlights the various markers used across different studies from 2016 to 2023, with a focus on muscle-invasive bladder cancer (MIBC) and non-muscle-invasive bladder cancer (NMIBC). Each study is represented with corresponding markers (squares), the number of patients analysed, and the subtypes identified in each study (circles). A key is provided at the bottom to decode the colours and markers used for classification [20,28,37,38,44,48,49,50,51,54,55,56,57,58,59,61,63,70,72,74,77,79,80,87,88,89,90].

**Table 1 diagnostics-14-02501-t001:** Molecular Classifications, Immunohistochemical Associations, and Therapeutic Strategies for Bladder Cancer Subtypes.

Molecular Classifications	Molecular Subtypes							
CMC	Ba/Sq		LumP	LumNS		LumUS	Stroma-rich	NE
TCGA	Ba/Sq		Luminal	Luminal		Luminal	B/Sq	NE
Lund	Ba/Sq	Uro B	Uro A			GU	Mes-like	Sc-NE
MDA	Basal		Luminal	Luminal	p53	Luminal	p53	
UNC	Basal		Luminal	Luminal		Luminal	Basal	Basal
Immunohistochemistry	KRT14, KRT5, CD44		Gata3, KRT20, Uroplakin2, Foxa1				Vimentin, desmin, SMA	Synaptophysin, chromogranin, CD56
Therapeutic strategy	Best sensitivity to NAC Response to ICI EGFR inhibitors		FGFR3 inhibitors Low sensitivity to NAC	Response to ICI Low sensitivity to NAC		Response to ICI Sensitivity NAC	Not specified	Response to ICI Combined chemotherapy

A comparison of bladder cancer molecular subtypes across major classification systems. It includes the associated immunohistochemical markers and recommended therapeutic strategies for each subtype. Ba/Sq (Basal/Squamous), CMC (Consensus molecular classification), EGFR (Epidermal Growth Factor Receptor), FGFR3 (Fibroblast Growth Factor Receptor 3), ICI (Immune checkpoint inhibitors), LumNS (Luminal Non-Specified), LumP (Luminal Papillary), LumUS (Luminal Unstable), MDA (MD Anderson Cancer Center), NAC (Neoadjuvant Chemotherapy) NE (Neuroendocrine) SMA (Smooth Muscle Actin, TCGA (The Cancer Genome Atlas), UNC (University of North Carolina).

**Table 2 diagnostics-14-02501-t002:** Summary of the main immunohistochemistry markers in molecular classification.

Molecular Subtypes	Immunohistochemistry Markers	Key Expression Markers
Luminal	GATA3, KRT20, Uroplakin2, FOXA1, HER2	High expression of GATA3 and KRT20; low levels of KRT5/6. Associated with urothelial differentiation
Basal/squamous (ba/sq)	KRT5/6, KRT14, p63, CD44	High expression of KRT5/6, KRT14, and p63
Genomically Unestable (GU)	p16, GATA3, HER2, CCDN1	High p16, GATA3, ERBB2, Low KRT5/6, KRT14. Associated with genomic instability and proliferation
Small cell/Neuroendocrine (Sc-Ne like)	Synaptpophysin, chromogranin A, CD56	Expression of neuroendocrine markers; aggressive phenotype and poor differentiation
Mesenchymal like (Mes like)	Vimentin, CDH1	High Vimentin, CDH1, Low GATA3, KRT5/6. Associated with epithelial-mesenchymal transitions; low expression of luminal and basal markers

Overview of the key immunohistochemistry markers used for molecular classification of bladder cancer subtypes. The table summarises the main molecular subtypes along with their associated immunohistochemistry markers and key expression characteristics.

## Data Availability

No new data were created or analysed in this study. Data sharing is not applicable to this article.

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
