# Peer review of "The Role of Immunohistochemistry as a Surrogate Marker in Molecular Subtyping and Classification of Bladder Cancer"

_diagnostics, 2024, doi:10.3390/diagnostics14222501_

Round 1

Reviewer 1 Report

Comments and Suggestions for Authors

Thank you for the invitation from the editor, which allowed me to read this manuscript at the first opportunity! The authors have invested a significant amount of effort in this work, presenting a detailed and logical introduction to the development of molecular subtyping in bladder cancer (BC) and providing a thorough summary of various molecular classification schemes. This work is highly valuable for urologists and related scientific researchers. Below are my specific comments on the manuscript:

  1. Bladder cancer (BC) is a highly heterogeneous disease, and determining the molecular subtype of patients is essential for achieving personalized treatment. Given the high costs and complexity of genomic techniques, the authors explore the role of immunohistochemistry (IHC) as a relatively economical surrogate marker for molecular subtyping in BC, which is a good choice for the topic! However, there are certain limitations to using IHC for differentiating BC subtypes. As the authors mentioned, BC is a highly heterogeneous disease, and IHC detection relies on the presence of positive cells in the tissue section, which has a high false-negative rate. For instance, HER2 expression is often scattered and does not form distinct clusters, making IHC detection prone to false negatives.

  2. The article is based on current research and literature reviews, citing various studies to provide detailed background and future research directions for the application of IHC as a tool for molecular subtyping in bladder cancer. The article highlights the potential of immunohistochemistry as a molecular marker, especially in clinical settings lacking complex genomic testing equipment. IHC could become an economically viable and easily implementable tool to replace costly and complex genomic techniques.

  3. The ultimate goal of molecular subtyping is to guide the treatment of BC patients. Although the article thoroughly discusses the clinical applications of molecular subtyping and IHC, it does not address the practical value of each molecular subtype. I suggest that the authors create a table to compare and analyze the application value of various molecular subtypes, which would help provide further research directions for readers.

  4. While the article mentions the potential of IHC, there is limited discussion on how to implement IHC in practice, how to choose markers, and the challenges faced in practical operations. This limits its utility in guiding clinical practice. Furthermore, there are significant differences in the antibodies used and the interpretation of results across different centers, leading to a lack of standardized criteria. I look forward to the development of a global standard for IHC interpretation in the future.

Author Response

The role of immunohistochemistry as a surrogate marker in molecular subtyping and classification of bladder cancer

Manuscript ID: diagnostics-3281320

Reviewer 1

  1. Bladder cancer (BC) is a highly heterogeneous disease, and determining the molecular subtype of patients is essential for achieving personalized treatment. Given the high costs and complexity of genomic techniques, the authors explore the role of immunohistochemistry (IHC) as a relatively economical surrogate marker for molecular subtyping in BC, which is a good choice for the topic! However, there are certain limitations to using IHC for differentiating BC subtypes. As the authors mentioned, BC is a highly heterogeneous disease, and IHC detection relies on the presence of positive cells in the tissue section, which has a high false-negative rate. For instance, HER2 expression is often scattered and does not form distinct clusters, making IHC detection prone to false negatives.

Response: We are in agreement with the reviewer regarding the challenges of using HER2 as an immunohistochemical (IHC) marker, given its heterogeneous expression in bladder cancer, which may lead to false-negative results. In the article, we emphasize the importance of understanding these limitations, especially when applied to Luminal Unstable subtypes, as characterized in the TCGA classification and referenced studies on Genomically Unstable (GU) and Luminal Unstable categories. We acknowledge that scattered HER2 expression complicates its application as a definitive marker, and as noted, standardizing protocols for IHC interpretation remains crucial. To address this, we have included studies and classifications that reflect HER2’s variable role, underscoring the need for further research to validate and optimize IHC marker panels, including HER2, to enhance their reliability and clinical relevance in molecular subtyping

  1. The article is based on current research and literature reviews, citing various studies to provide detailed background and future research directions for the application of IHC as a tool for molecular subtyping in bladder cancer. The article highlights the potential of immunohistochemistry as a molecular marker, especially in clinical settings lacking complex genomic testing equipment. IHC could become an economically viable and easily implementable tool to replace costly and complex genomic techniques.

Response: We appreciate the reviewer’s positive comments on our review. Thank you for recognizing our efforts to highlight immunohistochemistry (IHC) as a practical alternative to molecular techniques for subtyping bladder cancer, especially in settings with limited access to genomic testing. Our review aims to emphasize IHC's potential as a cost-effective tool to make molecular subtyping more accesible.

  1. The ultimate goal of molecular subtyping is to guide the treatment of BC patients. Although the article thoroughly discusses the clinical applications of molecular subtyping and IHC, it does not address the practical value of each molecular subtype. I suggest that the authors create a table to compare and analyze the application value of various molecular subtypes, which would help provide further research directions for readers.

Response: We appreciate the reviewer’s suggestion. In line with this recommendation, we have added a new table (Table 1) and developed a supporting paragraph based on a recent analysis by López-Beltrán et al. [23] that provides a comparative overview of molecular subtypes across the main classification systems. This table includes key immunohistochemical markers and therapeutic strategies associated with each subtype, giving a clear perspective on their clinical value and application in treatment decision-making. Additionally, we have highlighted the changes in yellow to draw attention to this new addition and facilitate its location within the manuscript (P8 L336-P9 L351).

  1. While the article mentions the potential of IHC, there is limited discussion on how to implement IHC in practice, how to choose markers, and the challenges faced in practical operations. This limits its utility in guiding clinical practice. Furthermore, there are significant differences in the antibodies used and the interpretation of results across different centers, leading to a lack of standardized criteria. I look forward to the development of a global standard for IHC interpretation in the future.

Response: We agree with the reviewer’s perspective on the need for practical guidance regarding IHC implementation in clinical practice. In response, we have expanded our discussion to cover the selection of specific markers, the operational challenges faced in routine settings, and the current variability in antibody use and result interpretation across centers. This review aims to address the complexities in IHC standardization and reinforces the need for a unified approach to improve its consistency and reliability. We also acknowledge the importance of developing a global standard for IHC interpretation to improve the utility in guiding molecular subtyping of bladder cancer.

Reviewer 2 Report

Comments and Suggestions for Authors

Congratulations to the authors, this is an excellent review! Read with pleasure and great interest. Only minor issues:

P2: Please eliminate discrepancy between L53 "NMIBC [...] affecting about 75%..." and L57 "remaining 30%". 

P3 L105 and P6 L214: "little progress". Please reconsider, there has been tremendous progress recently (immunotherapy, ADCs), ref 21 not up to date.

P5 L176-78: "low-grade papillary urothelial carcinoma [...] is associated with urothelial hyperplasia as a premalignant lesion". This is somewhat abridged or misrepresented, ref 22 obsolete. Hyperplasia (by definition) cannot be premalignant. 

P5 L178-80 "high-grade urothelial carcinoma [...] is associated with urothelial dysplasia as a premalignant lesion". Vide supra, please reconsider with current reference (WHO 5th ed 2022). Several pathways can lead to HGUC.

P9 L323: "UNC, MDA, TCGA..." abbreviations have not been introduced, please spell out in the figure caption for reader friendliness.

P9 L337-38: Additionally, when there are histomorphologically distinct regions within a single tumour, the molecular subtypes of these regions are often discordant." Please provide an appropriate reference for this statement. The whole paragraph (L329-341) comes without references.

P9 L342: "The recent introduction of the novel NanoString technology..." and L344 "the new NanoString technology" Please reconsider (20th anniversary of the company).

P9 L33-45 "91 MIBC patients (T2-T3) and NMIBC (Ta-T1)". Sounds as if 91 patients had MIBC. Please reconsider for clarity.

P10 L375-77: "Conventional pathological parameters, such as tumour stage and grade, have limited ability to predict the heterogeneous behaviour of BC, especially for tumours with similar stage and grade [38]." Ref 38 is not appropriate for this statement.

P10 L377-79 "IHC analysis [...] is crucial for the accurate diagnosis of BC [16]." This statement sounds rather strong and should be toned down a bit. IHC is really not necessary for an accurate diagnosis in the majority of cases.

P12 L512 "[36,52-63]" Ref 59 (Eckstein M et al.) is not IHC-based and should be taken out from the supplementary table, too.

P13 L517-18 ", respectively, in BC showing morphological features of urothelial differentiation [35,36,54-57,60]." I don´t understand this part of the sentence. Please consider splitting in two sentences.

P13 L555-56 "positive for p53 (p53-WT)". Please check for inconsistency with the originial paper. The p53-wildtype pattern is not just "positive" but represents a weak staining (defined as less than 50% of tumour cells in that paper) as a surrogate for a non-mutated TP53-Gene.

P13 L563 "[34]" wrong reference, should be Koll et al. [36].

P14 L568-9 "High concordance was observed between primary tumours and lymph node metastases [74]." Please discuss interesting discrepancy between refs 41 and 74 (here or on P11 L438).

P14 L587-589 "identified two more categories based on IHC and RT-PCR: the DP category (KRT20+, KRT5/6+) and the DN category (KRT20-, KRT5/6-)." These categories were defined by RT-PCR and found superior in comparison to IHC in ref 78, please clarify.

P15 L624-26 "Breyer J et al. focused on the expression of KRT20 (luminal) and KRT5 (basal) mRNA to subtype NMIBC, with IHC also used, showing that high KRT20 and low KRT5 levels were associated with worse prognosis [78]..." Vide supra, PCR found superior to IHC in that study, please specify.

P16 Please add reference numbers to the figure, e.g. Barth et al. [88]), difficult to correlate with references in the present form. Please remove Eckstein et al. (2018) whic is an exclusively PCR-based study.

P17 L675 "formalin fixation" not really a limitation, rather interesting potential (unlimited archived material).

Further literature for consideration:

Lerner et al. https://pubmed.ncbi.nlm.nih.gov/27376123/ 

Haghayeghi https://pubmed.ncbi.nlm.nih.gov/34218288/

Editing:

P5 L191: "P63" > p63

P10 L390 "immunohitochemistry" > immunohistochemistry

P18 L716 "1. 1. IARC. Bladder..."

P18 L718 "2. 2. Knowles MA..."

P18 L720 "3. 3. Stein JP..."

P18 L717, L719, L721, L725: dysfunctional hyperlink 

Author Response

The role of immunohistochemistry as a surrogate marker in molecular subtyping and classification of bladder cancer

Manuscript ID: diagnostics-3281320

Reviewer 2:

Comment 1: P2: Please eliminate discrepancy between L53 "NMIBC [...] affecting about 75%..." and L57 "remaining 30%". 

Response 1: We acknowledge the error and have corrected it by ensuring consistency in the percentages cited. The revised text on P2, L53, and L57 accurately reflects the proportions highlighted in yellow. Thank you for noting this

Comment 2: P3 L105 and P6 L214: "little progress". Please reconsider, there has been tremendous progress recently (immunotherapy, ADCs), ref 21 not up to date.

Response 2: Thank you for your valuable feedback. As per your suggestion, we have updated the manuscript on P3, L105-08, and P6, L220-22 to reflect recent advances in bladder cancer treatment, including immunotherapy and antibody-drug conjugates. We have revised the statement regarding "little progress" and replaced it with a more accurate description of recent developments in the field. Additionally, we have replaced the outdated reference [21] with the following updated citation: Gill E, Perks CM. Mini-Review: Current Bladder Cancer Treatment—The Need for Improvement. Int J Mol Sci. 2024 Jan 26;25(3):1557. http://doi.org/10.3390/ijms25031557.

Comment 3: P5 L176-78: "low-grade papillary urothelial carcinoma [...] is associated with urothelial hyperplasia as a premalignant lesion". This is somewhat abridged or misrepresented, ref 22 obsolete. Hyperplasia (by definition) cannot be premalignant. 

Response 3: We thank the reviewer for the comment. Following the reviewer’s observation and recent sources, including the WHO classification (5th ed. 2022), we have revised the text on P5, L177-182, to clarify that low-grade papillary urothelial carcinoma is commonly associated with early genetic alterations, such as deletions in chromosome 9 and FGFR3 mutations, suggesting a clonal link with hyperplastic precursors but not indicating hyperplasia as premalignant. We have also updated and modified reference 23 to reflect this understanding.

Comment 4: P5 L178-80 "high-grade urothelial carcinoma [...] is associated with urothelial dysplasia as a premalignant lesion". Vide supra, please reconsider with current reference (WHO 5th ed 2022). Several pathways can lead to HGUC.

Response 4: We thank the reviewer for the comment.We have revised the text on P5, L182-186, to acknowledge the multiple pathways leading to high-grade urothelial carcinoma (HGUC), which can include flat urothelial dysplasia and carcinoma in situ (CIS) as precursors that share molecular characteristics with invasive carcinoma. According to recent guidelines (WHO 5th ed. 2022), we recognize that HGUC is associated with a range of molecular alterations, including mutations in TP53, RB1, and PTEN, and that dysplasia or CIS can sometimes serve as early indicators rather than direct precursors. We have also updated reference 23 in line with this broader understanding of HGUC pathogenesis.

Comment 5: P9 L323: "UNC, MDA, TCGA..." abbreviations have not been introduced, please spell out in the figure caption for reader friendliness.

Response 5: We have made the requested modifications to the figure caption, spelling out the abbreviations for clarity and reader friendliness. As outlined in the paragraph on P10, L 399-401, the updated text now includes the full names: University of North Carolina (UNC), MD Anderson Cancer Center (MDA), The Cancer Genome Atlas (TCGA), Consensus Molecular Classification (CMC), and Lund University (Lund).

Comment 6: P9 L337-38: Additionally, when there are histomorphologically distinct regions within a single tumour, the molecular subtypes of these regions are often discordant." Please provide an appropriate reference for this statement. The whole paragraph (L329-341) comes without references.

Response 6: We have now added the appropriate references to the paragraph in question (P9 L365), including reference [12] and new reference [34].

Comment 7: P9 L342: "The recent introduction of the novel NanoString technology..." and L344 "the new NanoString technology" Please reconsider (20th anniversary of the company).

Response 7: We are in complete agreement with the reviewer’s observations. We have revised the wording in the manuscript to reflect that NanoString technology has been established for nearly two decades, rather than describing it as "recent" or "novel." The updated text now emphasizes the application of NanoString in bladder cancer molecular subtyping. P9 L366-367, 372 (highlighted in yellow).

Comment 8: P9 L33-45 "91 MIBC patients (T2-T3) and NMIBC (Ta-T1)". Sounds as if 91 patients had MIBC. Please reconsider for clarity

Response 8: We have revised the text for clarity. It now specifies the cohort composition, distinguishing between MIBC (T2-T3) and NMIBC (Ta-T1) within the 91 patients. P9 L368-370 (highlighted in yellow).

Comment 9: P10 L375-77: "Conventional pathological parameters, such as tumour stage and grade, have limited ability to predict the heterogeneous behaviour of BC, especially for tumours with similar stage and grade [38]." Ref 38 is not appropriate for this statement.

Response 9: We thank the reviewer for the comment. We have removed the previous Reference [38] and updated it to Reference [23], which more accurately aligns with our statement regarding the limited predictive power of conventional parameters, such as tumor stage and grade, in assessing the heterogeneous behavior of bladder cancer (P11 L412).

Comment 10: P10 L377-79 "IHC analysis [...] is crucial for the accurate diagnosis of BC [16]." This statement sounds rather strong and should be toned down a bit. IHC is really not necessary for an accurate diagnosis in the majority of cases.

Response 10: Thank you for the suggestion. We have revised the text on P11 L413 (highlighted in yellow) to read: "In this context, IHC analysis of biomarker expression complements the morphological evaluation of the tumor, offering information to support the accurate diagnosis of BC." This updated phrasing reflects a more balanced emphasis on the role of IHC in diagnosis.

Comment 11: P12 L512 "[36,52-63]" Ref 59 (Eckstein M et al.) is not IHC-based and should be taken out from the supplementary table, too.

Response 11: Thank you for pointing this out. We have removed the reference to Eckstein M et al. from the manuscript’s main references, the supplementary material, and Figure 4.

Comment 12: P13 L517-18 ", respectively, in BC showing morphological features of urothelial differentiation [35,36,54-57,60]." I don´t understand this part of the sentence. Please consider splitting in two sentences

Response 12: Thank you for your feedback. We have revised the text as per your suggestion, splitting the sentence for improved clarity. The modified statement now appears on P14 L552-553, where it is highlighted in yellow.

Comment 13: P13 L555-56 "positive for p53 (p53-WT)". Please check for inconsistency with the originial paper. The p53-wildtype pattern is not just "positive" but represents a weak staining (defined as less than 50% of tumour cells in that paper) as a surrogate for a non-mutated TP53-Gene

Response 13: We thank the reviewer for this helpful comment. In the study by Helal et al., the p53-WT subtype is indeed characterized by weak nuclear staining in less than 50% of tumor cells, serving as a surrogate for a non-mutated TP53 gene. We have revised the manuscript to reflect this, with changes on P14, L590-591, highlighted for clarity.

Comment 14: P13 L563 "[34]" wrong reference, should be Koll et al. [36].

Response 14: Thank you for noting this discrepancy. We have corrected the reference as suggested, replacing [34] with Koll et al. [38] on P14 L598, as highlighted in the updated manuscript.

Comment 15: P14 L568-9 "High concordance was observed between primary tumours and lymph node metastases [74]." Please discuss interesting discrepancy between refs 41 and 74 (here or on P11 L438).

Response 15: Thank you for your valuable suggestion to discuss the discrepancies between references 42 and 74 regarding the concordance between primary tumors and lymph node metastases. We have added a detailed comparison to the manuscript, addressing the methodological differences and their implications for understanding phenotypic adaptation and intratumoral heterogeneity in the Ba/Sq subtype. This addition, highlighted in yellow on P14 L604 and P15 L605-608, has enriched the discussion, particularly as limited studies are exploring molecular subtypes and their correlation with lymph node metastasis. We appreciate your insightful recommendation, as it highlights an important area of investigation in metastatic bladder cancer.

Comment 16: P14 L587-589 "identified two more categories based on IHC and RT-PCR: the DP category (KRT20+, KRT5/6+) and the DN category (KRT20-, KRT5/6-)." These categories were defined by RT-PCR and found superior in comparison to IHC in ref 78, please clarify.

Response 16: We thank the reviewer for the comment. Based on Breyer et al. [78], we have updated the manuscript to specify that these categories—DP (KRT20+, KRT5/6+) and DN (KRT20-, KRT5/6-)—were identified primarily through RT-PCR, which the authors demonstrated to be superior to IHC alone in predicting prognosis in NMIBC.

We revised the text to emphasize that while luminal-basal categorization by IHC remains a valuable approach, RT-PCR provides a more refined classification, capturing distinctions in KRT20 and KRT5 expression with greater prognostic accuracy. This addition highlights the complementary role of RT-PCR alongside IHC, especially for prognostic stratification within these subtypes. We have noted this clarification in the manuscript, highlighted in yellow on P15 L626-632.

Comment 17: P15 L624-26 "Breyer J et al. focused on the expression of KRT20 (luminal) and KRT5 (basal) mRNA to subtype NMIBC, with IHC also used, showing that high KRT20 and low KRT5 levels were associated with worse prognosis [78]..." Vide supra, PCR found superior to IHC in that study, please specify.

Response 17: We thank the reviewer for the comment. Continuing from the previous clarification, we have specified in the revised text on P16 L669-670 that Breyer et al. demonstrated the superior accuracy of RT-PCR over IHC alone in determining KRT20 and KRT5 expression for NMIBC prognosis.

Comment 18: P16 Please add reference numbers to the figure, e.g. Barth et al. [88]), difficult to correlate with references in the present form. Please remove Eckstein et al. (2018) which is an exclusively PCR-based study.

Response 18: We thank the reviewer for the comment. We have updated Figure 4 by adding reference numbers to each study within the figure for easier correlation with the reference list and have removed the study by Eckstein et al. (2018), as it is exclusively PCR-based. We appreciate your guidance in enhancing the clarity and scientific rigor of our manuscript.

Comment 19: P17 L675 "formalin fixation" not really a limitation, rather interesting potential (unlimited archived material).

Response 19: We thank the reviewer for the insightful comment regarding formalin fixation. Initially, we included this as a limitation due to the variability in fixation times for archived samples, as prolonged fixation can sometimes affect the quality of immunohistochemical studies by altering antibody reactivity. However, we understand that archived material also offers significant potential for retrospective studies. Following the reviewer's suggestion, we have removed formalin fixation from the limitations section to reflect this more positive perspective. We have specified in the revised text on P18 L718.

Further literature for consideration:

Thank you for the suggestions regarding additional literature. We have reviewed both articles: Lerner et al. This article has been incorporated into our review, particularly adding valuable context to the consensus regarding the basal-squamous subtype in bladder cancer. We have referenced this on P7 L273-280 as [29] as suggested by the reviewer. Haghayeghi et al. This reference was already included in our manuscript as reference [61].

Editing: We thank the reviewer for their attention to detail. All specified editing changes have been correctly implemented:

  • P5 L191: Corrected "P63" to "p63."
  • P10 L390: Corrected "immunohitochemistry" to "immunohistochemistry."
  • P18 L716: Corrected citation numbering format from "1. 1. IARC. Bladder..." to "1. IARC. Bladder..."
  • P18 L718: Corrected citation numbering format from "2. 2. Knowles MA..." to "2. Knowles MA..."
  • P18 L720: Corrected citation numbering format from "3. 3. Stein JP..." to "3. Stein JP..."
  • P18 L717, L719, L721, L725: Addressed dysfunctional hyperlinks, ensuring each link functions as intended.

Thank you for these recommendations; all adjustments have been thoroughly reviewed and applied.

We would like to express our sincere gratitude to the reviewer for their valuable comments and insightful suggestions throughout this process. It has been a true pleasure to work on these revisions, as the reviewer’s feedback not only enhanced the quality of our manuscript but also contributed significantly to our own learning and development as authors. We deeply appreciate the thoroughness of the review, and we believe that, thanks to these contributions, our work will offer meaningful insights to the scientific community, especially for pathologists and urologists specializing in bladder cancer.

Thank you again for this opportunity and for your support.
